# Shaving Technique and Compression Therapy for Elephantiasis Nostras Verrucosa (Lymphostatic Verrucosis) of Forefeet and Toes in End-Stage Primary Lymphedema: A 5 Year Follow-Up Study in 28 Patients and a Review of the Literature

**DOI:** 10.3390/jcm9103139

**Published:** 2020-09-28

**Authors:** Robert J. Damstra, Janine L. Dickinson-Blok, Harry G.J.M. Voesten

**Affiliations:** 1Member European Reference Network ERN (VASCERN PPL), Department of Dermatology, Expert Center for Lymphovascular Medicine, Nij Smellinghe Hospital, 9202 NN Drachten, The Netherlands; j.dickinson@nijsmellinghe.nl; 2Department of Vascular Surgery, Nij Smellinghe Hospital, 9202 NN Drachten, The Netherlands; hgjvoesten@me.com

**Keywords:** primary lymphedema, elephantiasis nostras verrucosis, surgery, compression, shaving, garment, toecaps, lymphostatic verrucosis

## Abstract

Background. Longstanding lymphedema can lead, especially when there is recurrence of erysipelas, to irreversible elephantiasis nostras verrucosa (ENV). This predisposes to new episodes of erysipelas, leading to further damage of the lymphatics and deterioration of the lymphedema as a whole. We report the results of 28 patients with primary lymphedema and surgical removal ENV of the forefoot and toes treated between 2006 and 2014. Method: Retrospective descriptive 5 year follow-up study of 28 patients with various diagnosis of primary lymphedema. Wound healing time, number of erysipelas, body mass index (BMI), recurrence of EVN and types of compression were documented during follow-up. Results: After preoperative multidisciplinary work up, operation of the toes with shaving and excision was performed within a conservative treatment program. During the follow up, the number of erysipelas attacks decreased dramatically (mean 17.6 vs. 0.6). Before treatment, no toecaps were used; and in follow up, it was a part of treatment. Recurrence of ENV was not observed. Compared to the literature with often BMI > 35, the mean BMI in our group was 30.0 (overweight). In 12 patients, we concurrently performed circumferential suction-assisted lipectomy for end-stage lymphedema of the leg. Conclusion: Although lymphedema patients are treated with garments during the maintenance phase, compression of the toes is often too challenging. Surgical removal of the verrucosis of toes is an effective therapeutic modality as part of an integrated lymphedema treatment program to restore the shape of the toes and enable the wearing of toecaps. This technique can also be effective for ENV of origins other than primary lymphedema. Although ENV is a generally accepted term, it can have undesirable connotations. We suggest using a more inclusive name such as lymphostatic verrucosis, because long-lasting lymphatic impairment is involved in all ENV and the term verrucosis is above discussion.

## 1. Introduction

Elephantiasis nostras verrucosa (ENV) represents cutaneous changes with fibrosis, papillomatosis, hyperkeratosis and verrucosis due to long-lasting lymphatic stasis, bacterial and mycological colonization, recurrence of infections and sometimes in combination with chronic venous insufficiency or severe obesity [1]. ENV is a manifestation of end-stage chronic lymphatic stasis occurring in for instance end-stage filariasis and podoconiosis [2], as they are morphologically similar but the etiology of the lymphatic stasis differs.

Swelling and thickening of the longitudinal skinfold of the toe was first described in 1887 by Moritz Kaposi and later in 1976 by Stemmer [3] as an early clinical sign of lymphedema and is called the Kaposi–Stemmer sign. ENV characterizes the end stage of lymphatic impairment, with profound dermal and soft tissue changes. According to the International Society of Lymphology, elephantiasis is classified as Stage 3 lymphedema (in a Stage 0 to 3 clinical classification of lymphedema) and is characterized by trophic skin changes such as acanthosis, further deposition of fat and fibrosis, and warty overgrowths with or without edema [4].

Elephantiasis leads to limb disfigurement, recurrent episodes of erysipelas and further deterioration of lymphedema when left untreated. This phenomenon is often seen in (non)filarial lymphedema. Davis [5] first published his work on the lymphatics in ENV in 1955 and called it lymphostatic verrucosis.

In 1969, Castellani et al. [6] reported on elephantiasis tropica and its relation with recurrent episodes of erysipelas (lymphangitis recurrens elephantogenica) in filariasis. He recognized four subtypes of elephantiasis related to the origin of lymphatic stasis:(1)elephantiasis tropica (due to filariasis);(2)elephantiasis nostras due to recurrent infections;(3)elephantiasis symptomatica due to other conditions such as surgery, podoconiosis [7], tuberculosis, and overweight [8];(4)Elephantiasis due to primary lymphedema.

This classification is based on the etiology of lymphostasis. The clinical signs of ENV are the distinctive dermatological findings of severe epidermal and dermal changes, with dermal fibrosis, hyperkeratotic, verrucosis, papillomatous, nodular lesions displaying a cobblestone-like appearance. This warty skin appearance serves as a source of bacterial and fungal colonization. Finally, skin abrasions and a moist environment cause recurrent cellulitis/infections, leading to progression of lymphedema due to further disruption of lymphatics.

The terms cellulitis/erysipelas/infection are frequently used but officially describe separate entities. Cellulitis is bacterial infection of the inner layers of the skin often caused by *Streptococci* and *Staphylococcus aureus*. Erysipelas is a bacterial infection involving the more superficial layers of the skin and caused by *Streptococci*. Infection is a general term for invasion of the body by a pathogen.

All patients with ENV have a long history of limb swelling, with sometimes (local) treatment for many years, which is often complicated by secondary infection/cellulitis, slowly progressing to this debilitating and disfiguring chronic manifestation. While the legs are treated with conservative measurements, including specialized long-lasting compression therapy, often the toes are left untreated, as it is technically difficult.

Dean et al. [8] described 21 patients with ENV exclusively related to obesity, soft tissue infections and chronic venous insufficiency. They suggested that ENV could complicate lymphedema. If initial lymphedema is combined with morbid obesity, ENV could appear in an early stage.

As the largest cause of lymphedema globally is lymphatic filariasis, elephantiasis is mostly caused by advanced lymphedema due to this infection [9]. Nonfilarial lymphedema such as podoconiosis is a common but neglected tropical disease in certain regions of the world [10] Podoconiosis is a noninfectious geochemical elephantiasis caused by exposure of bare feet to irritant alkalic clay soils in both men and women. It is found in at least 10 low-income countries in tropical Africa, Central America and northwest India, where such soils coexist with high altitude and high seasonal rainfall. When untreated, ENV-like lesions are often seen on the toes, feet, and lower legs with nodule formation [7].

Primary lymphedema can also progress to the condition of elephantiasis if left untreated [11], and infection is allowed to supervene. Alternative terms for this disorder include lymphostatic verrucosis, lymphostatic papillomatosis cutis, elephantiasis crurum papillaris et verrucosa and mossy foot and/or leg.

In general, prevention of swelling of the toes and therefore prevention of ENV can be achieved by timely compression and the use of a toecap, which is a type of knitted compression garment dedicated for the toes and forefoot, comparable with a glove for the hand. This advanced technique for the toes is not widely used or available.

When ENV occurs together with hyperkeratotic lesions in end-stage lymphedema, some studies show the efficacy of oral retinoid therapy [12,13] in flattening hyperkeratotic, verrucous and papillomatous lesions. Etretinate has more recently been replaced by acitretin, which has a similar potency with less severe side effects [14,15].

When irreversible skin changes such as fibrosis and nodules occur, removal of pathological tissue may be indicated by electrocauterization [14] or surgical excision [15]. Postoperative treatment includes compression therapy in addition to wound care.

To treat ENV of the toes and upper foot specifically, we used a shaving technique with a razor blade and electrocauterization in a similar manner to that used in rhinophyma. Lazzeri et al. [16] studied the shaving and excision technique to remove hypertrophic tissue while sparing the sebaceous glands and hair follicles for reepithelization and restored a more natural configuration of the nose.

In ENV, this same technique may be used where hair follicles are spared to allow for re-epithelization of the skin of the toes. In this study, we report our results using this shaving technique in lymphedema patients with protocolized postoperative wound care and compression therapy in a multidisciplinary setting for a minimum of 5 years and review the current literature on this subject. Additionally, the results of genetic testing on these patients are described.

## 2. Patients and Methods

### 2.1. Patients

Patients selected for this study were referred to the Expert Center for Lymphovascular Medicine of the Nij Smellinghe Hospital in Drachten for treatment for lymphedema and concomitant ENV between 2006 and 2014. All patients provided informed consent in accordance with local guidelines. All patients (13 males, 15 females) were diagnosed with primary lymphedema. The main indication for operation was severe recurrent attacks of erysipelas caused by loss of skin integrity/severe verrucous skin deformation/impairment of locomotion by the deformities after unsuccessful multidisciplinary conservative treatment and the inability to wear effective garments and toecaps.

All patients were physically fit to undergo surgery. Preoperatively, surgical and anesthesiologic work up were completed, and full compliance was obtained to wear garments and toecaps for the rest of their life.

### 2.2. Measurements

All patients were measured for weight, body mass index (BMI), number of erysipelas, and type of compression and their legs and toes were photographed initially and during follow up.

In addition, data were collected at several time points—t = 0; t = 3 weeks; t = 3 months; t = 1 year—and then their medical history was taken annually, focusing on functionality of the leg and foot. Patients were questioned by phone or life consultation to obtain the accurate data on medical status and function (t = 5 years).

### 2.3. Gene Testing

Gene testing was performed under strict indications. Studies have shown that gene identification is most successful when there are signs of a syndrome, concomitant clinical features, early onset of lymphedema (<1 year) or a positive family history [17]. In our patient population, we performed gene testing when lymphedema appeared before 10 years of age. In patients with a history of lymphedema developing after the age of 10 years, the diagnosis was made based on clinical phenotyping.

We initially started with gene testing for two genes (*FLT4* and *FOXC2*), and since 2014, we have performed next-generation sequencing (NGS). The final diagnosis, ORPHAnet codes and results of genetic testing are presented in Table 1.

### 2.4. Preoperative Treatment

Patients underwent multidisciplinary, conservative treatment for 3–7 days preoperatively to remove the pitting component of the leg and toes and to optimize skin care. This treatment consisted of short-stretch compression bandaging of the leg and toes, exercise, skin care and self-management instructions. No additional manual lymph drainage was performed.

### 2.5. Surgical Procedure

Patients were given both general or regional anesthesia depending on whether the procedure was performed as a single procedure or combined with a circumferential suction-assisted lipectomy (CSAL) [18]. Cefazolin was given intravenously at surgery induction and repeated if surgery lasted longer than 2 h. Afterwards, no further antibiotics were given according to the standard stewardship offered by the hospital microbiologist. In 12 cases, the shaving procedure was combined with CSAL. In these cases of combined treatment, a superwet tumescent technique was used for the leg in combination with general anesthesia to allow for a prolonged operating time.

When shaving alone was performed, regional epidural anesthesia was used. The ENV was removed surgically with a razor blade until pinpoint bleeding could be observed (see Figure 1). Hemostasis was obtained by topically applying gauze drenched in saline/adrenaline solution. Diathermy was used only as a last resort to coagulate persistent vessels, and we did not use the electric knife for excision. In case the deformity was unsuitable for removal by razor blade (for instance, syndactyly) because of its magnitude, additional surgical excision by knife and suturing was performed (see Figure 2 and Figure 3).

### 2.6. Postoperative Treatment

In the surgical theater, the wounds were covered with nonadhesive analgesic silicone wound dressings and covered with compression bandages of the foot (and leg when CSAL was performed). The toes were separately bandaged with Elastomull Haft ^®^ (BSN Medical GmbH, Hamburg, Germany), and the leg up to the groin by specially trained staff with multilayer short-stretch compression bandages from the Trico^®^ IMCB system (BSN Medical GmbH, Hamburg, Germany) consisting of an initial protective layer (Tricofix^®^, BSN Medical GmbH, Hamburg, Germany) covered with two layers of synthetic cast wadding (Delta-Rol^®^ S, BSN Medical GmbH, Hamburg, Germany). Two layers of Trico^®^ 12 cm × 4 m bandaging material were applied over the synthetic cast wadding. The first change in bandages within 24 h by the specialized wound nurse followed by formal wound inspection was three days after surgery.

In the case of CSAL, measurements were taken by the dermatologist to order custom-made flat-knitted compression garments, and in a later stage, when the wounds of the foot and toes healed, measurements for additional toecaps were taken. Wound care was performed every two days. After 6 days, wound care was continued with toe bandaging, and the garment for the leg was put in place. The first days of mobilization weightbearing was allowed by the use of a special offloading “wedge” shoe to reduce painful flexing movements of the toes. In addition, standard painkillers were prescribed. The patient was discharged from the hospital as soon as the wounds were healed or the patients themselves or a homecare nurse were able to complete the final part of wound care. Additional circumferential liposuction was never a reason for longer hospitalization.

During the clinical phase, one garment and separate toecaps were made based on measurements. At the one-month visit, the hosiery was measured again. On a yearly basis, patients needed between two and four flat-knitted garments/toecaps (see Figure 2). The dermatologists took all measurements for compression materials themselves.

## 3. Results

The study population consisted of 28 patients with primary lymphedema (see Table 1). Their characteristics are summarized in Table 2.

During the study, six patients had a follow up of three years. All other patients (*n* = 22) had a full follow up of at least 5 years.

## 4. Discussion

This retrospective study in 28 patients with various forms of primary lymphedema and elephantiasis nostras verrucosa had a few clinical features in common: almost all had several episodes of threatening erysipelas, and none of them had previously had adequate toe compression. Although obesity is often reported to be a complicating factor in the literature, morbid obesity in our group was only present in 6/28 patients.

Based on the etiological classification from Castallani et al. [6], all our patients should be classified as having ENV 4 primary lymphedema.

To date, the cause of ENV in most patients has been reported as filariasis, morbid obesity or infection. This study is the first retrospective group of exclusively primary lymphedema patients, mostly complicated by recurrent infections and lack of proper toe compression.

In the publication from Castallani et al. [6], he recognized a subgroup ENV 2 (recurrent infections). We wondered whether the recurrence of infections was due to primary lymphedema in this group, as it was also classified in ENV 4. In approximately 80% of the cases without a known cause of erysipelas, lymphatic impairment seems to be the cause [19,20]. Consequently, after starting antibiotics in the initial phase, a thorough investigation to rule out previously not diagnosed lymphatic impairment should be carried out.

Therefore, close phenotyping, and, if necessary, genotyping, is essential in order to understand the cause of ENV and start concomitant treatment of the underlying lymphedema.

When studying literature on ENV, there are few studies on compression. From a lymphological point of view, compression is mandatory in the treatment of lymphedema in combination with lifestyle interventions, such as improving mobility and weight reduction/control [4,21,22].

In the last decade, more and better-quality toecaps have been produced to reduce the swelling of the toes. In our expert center, even in primary lymphedema in children, toecaps in addition to garments are prescribed at early signs of toe swelling to prevent irreversible complications such as erysipelas, fibrosis, papillomatosis and eventually ENV.

In this study, we demonstrated an important reduction in episodes of erysipelas after treatment of ENV based on surgery and a strict use of compression garments and toecaps in the follow up. Furthermore, additional attention for mobility, active lifestyle and weight reduction is important in lymphedema treatment in general and in this group ENV especially. Because most patients suffer from end-stage lymphedema with severe complications such as overweight and ENV, treating these aspects is a challenge.

Because end-stage ENV was combined with end-stage lymphedema of the leg in 12 patients, we performed additional CSAL concurrently. Although the surgical removal of ENV led to more pain and an increased need for painkillers due to wound healing, there was no interference or complications for swelling reduction in the leg by CSAL.

In the study by Dean et al. [8], he reviewed 21 patients with ENV 3 due to concomitant morbid obesity (19 patients > BMI 40) and chronic venous insufficiency. In our group, obesity was not the main objective (6/28 BMI > 35). All publications in the literature did not mention a follow up, and only a few mentioned compression therapy. Our study shows that, after treatment, compression therapy with garments and toecaps is essential.

In 2008, Sisto et al. [23] reviewed ENV. Interestingly, the lymphological component is mentioned for lymphatic obstruction but not a lymphatic overload by raised filtration due to infection, obesity and lack of mobility. In the treatment phase, compression is not a mandatory modality, and toecaps are not mentioned. The focus is mainly on histopathology, and they suggest that ENV is a progressive disease. We demonstrated that even with ENV as an end-stage condition, intensive treatment including shaving technique or excision, mandatory compression garments of the legs and toecaps, weight reduction and improvement of mobility can be helpful to reduce erysipelas attacks and stop recurrence of severe ENV after 5 years.

In our patient group, we found no patients with Milroy disease, although that is the most common gene defect (*FLT4*) known in primary lymphedema at this time [24]. Perhaps this is because Milroy disease only affects the lower limbs and often has a mild course. Furthermore, Milroy disease is often congenital, and therefore, patients are already treated from an early date with compression. In *FOXC2*, lymphatic impairment of the lower limbs is observed in combination with venous insufficiency, leading to a considerable lymphatic load and high numbers of infections.

As most patients had late-onset lymphedema, the concomitant recurrent erysipelas caused skin changes and consequently end-stage ENV. Early recognition of toe swelling and treatment with local compression can prevent infections and the formation of ENV. We showed that after ENV operation and toecaps, the number of erysipelas attacks decreased dramatically during follow up. Additionally, most patients maintained a stable weight (mean 30.0 vs. 30.6 after 5 years).

Finally, elephantiasis nostras verrucosa is more of a historic name and is seen as a stigma by those who are affected, and the term “nostras” refers to “nostro”, meaning “our”. Because long-lasting lymphatic impairment is involved in all ENV, we suggest using the term lymphostatic verrucosis used by Davis [5], which has no undesirable connotations.

In Table 3, we summarized the literature on ENV of the lower legs not related to tropical elephantiasis or podoconiosis or other regions of the body.

## 5. Conclusions

ENV is an end-stage condition with papillomatosis, verrucosis, and skin deformity in several conditions, such as tropical elephantiasis, after recurrent infections and is caused by obesity and immobility. We identified 28 ENV patients initially known with primary lymphedema, recurrent periods of erysipelas and obesity in 6 cases. All patients had no previous toe compression.

After treatment by shaving and/or excision in 12 cases concurrent with circumferential suction-assisted liposuction, patients received compression of the legs and toes, exercise and weight control in the maintenance phase. The number of erysipelas attacks decreased dramatically, and the ENV was under control with lifelong leg and toe compression.

Our review of the literature showed that follow up in most reports was not mentioned as there was no concomitant compression treatment. We advise a multidisciplinary approach with proper conservative preoperative treatment, operation, and good wound care, including compression of the legs and toes. During the maintenance phase, a lymphological approach is essential in order to maintain the operative result, reduce the number of erysipelas attacks and keep the lymphedema under control. Compression garments and toecaps, weight reduction and exercise will properly control lymphedema and operative results.

Although ENV is a generally accepted term, it can have undesirable connotations. We suggest using a more inclusive name such as lymphostatic verrucosis, because long-lasting lymphatic impairment is involved in all ENV and the term verrucosis is above discussion.

## Figures and Tables

**Figure 1 jcm-09-03139-f001:**
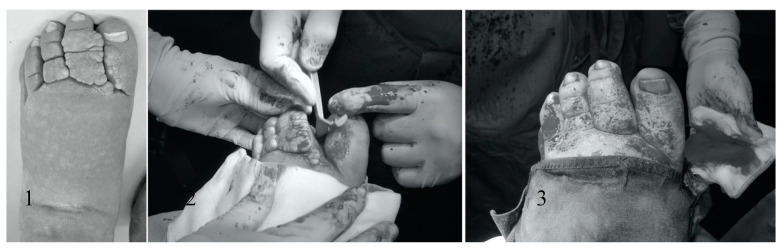
Procedure of shaving late-onset lymphedema left leg with razorblade: (**1**) preoperatively, (**2**) during procedure, and (**3**) end result postoperatively.

**Figure 2 jcm-09-03139-f002:**
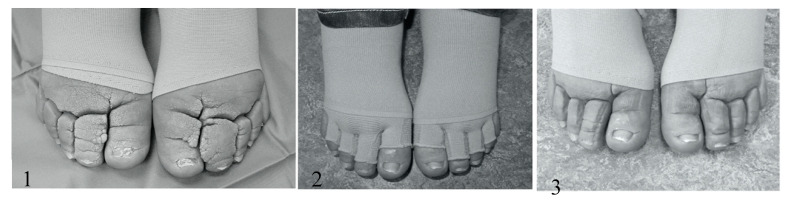
Procedure of excision and reconstruction in FOXC2 lymphedema: (**1**) preoperatively, (**2**) after 12 weeks with toecaps, and (**3**) result after 12 weeks.

**Figure 3 jcm-09-03139-f003:**
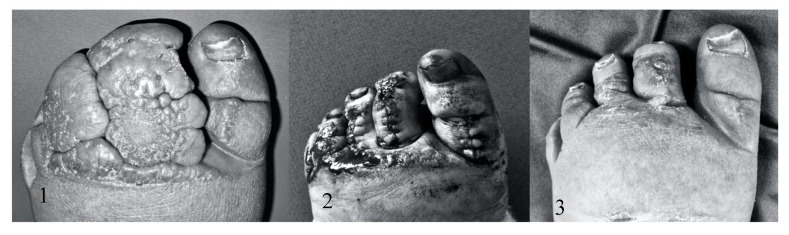
Procedure of excision and reconstruction in FOXC2 lymphedema: (**1**) preoperatively; (**2**) after 1 week and (**3**) after 12 weeks.

**Table 1 jcm-09-03139-t001:** Diagnosis of primary lymphedema and eventually corresponding gene defects.

Diagnosis of Primary Lymphedema	ORPHAnet Code	Genetic Mutation
Late-onset nonhereditary lymphedema	ORPHA 90185	Not tested: *n* = 13
		No *FLT4/FOXC2*: *n* = 5
		Results NGS:
		Negative: *n* = 1
		*FAT4*: *n* = 1
		CCA CHrX: NM000169.2 C457G > A (Asp 153 Asn); class 3 hetrozygote: *n* = 1
Lymphedema distichiasis syndrome	ORPHA 33001	*FOXC2* (*n* = 3)
Meige syndrome	ORPHA 90186	Not tested (*n* = 2)
Generalized lymphatic dysplasia	ORPHA 2136	No gene found (*n* = 1)
Noonan syndrome	ORPHA 648	*RIT1* gene (*n* = 1)

**Table 2 jcm-09-03139-t002:** Clinical data and results of 28 patients who underwent surgical treatment for elephantiasis nostras verrucosa.

	At Time of Admission	Follow-Up
Gender (male/female)	M = 13/F = 15	
Age of onset lymphedema	Δ 16.3 (0–34)	
Age first visit our clinic	Δ 44.7 (32–66)	
Duration of lymphedema at first visit in years	Δ 27.5 (6–36)	
Age at onset of swelling toes in years	Δ 21 (0–44)	
Duration of swelling toes at first visit in years	Δ 26.6 (5–50)	
Mean age (years) at first attack of erysipelas	Δ 20.5 (5–44)	
Wound healing in weeks	Δ 3.6	
Wearing of toecaps	*n* = 28/0	*n* = 28/28
N erysipelas:	Δ 17.6 (self-reported and hospital records)	Δ 0.6 (0–9) (during follow-up phase)
≤10	*n* = 10; Δ 3.3 (1–9)	
>10	*n* = 18; Δ 25.5 (10–50)	
Overweight/obesity	BMI < 28: *n* = 10	BMI < 28: *n* = 9
	BMI 28–35: *n* = 12	BMI 28–35: *n* = 13
	BMI > 35: *n* = 6	BMI >35: *n* = 6
Total mean:	Δ 30.0 (20.0–42.3)	Δ 30.7 (20.1–44.1)
Weight control during follow-up		Improved: *n* = 9
		Stable: *n* = 14
		Worsened: *n* = 5

**Table 3 jcm-09-03139-t003:** Literature on ENV of the legs.

Reference	Location of ENV	Diagnosis	N of Patients	Erysipelas n/year PreOP	Overweight/BMI	Type of Treatment	Compression/Toecaps	Follow-Up
Ferrandiz et al. [25]	Feet	EVN type 2 (infections)	1	Several	Unknown	Excision	Not mentioned	1 year
Sinha et al. [2]	Feet	EVN type 3: cutaneous tuberculosis	1	NA	Normal according to picture	Medication	Not mentioned	No
deGodoy et al. [11]	Lower leg	EVN type 4	2	NA	Unknown	Excision	Yes	No
Yang et al. [26]	Lower leg	EVN type 2	1	Several	Unknown	Etretinate	Not mentioned	No
Kar Keong et al. [27]	Lower leg	EVN type 4	1	6x/2 years	Obese	Suggestion for amputation	Not mentioned	
Borst et al. [28]	Lower leg	EVN type 3	1	Several	Unknown	Surgical debridement and maggots	Not mentioned	No
Liaw et al. [29]	Lower leg	EVN type 3 (incl. obesity)	1	NA	BMI 40.4	Passed away	Not mentioned	No
Dean et al. [8]	Lower leg	EVN type 3 (mainly obesity and CVI)	21	Many times	BMI 30–40 (*n* = 2) BMI > 40 (*n* = 19)	Retinoids, debridement, and excision	Yes	No
Schiavo et al. [30]	Lower leg	EVN 3 (obesity and CVI)	1	NA	BMI 41	Conservative without result	Yes	No
Hennessy et al. [31]	Lower leg	EVN 3 (incl. obesity)	1	NA	BMI 38.8	Not mentioned	Yes	No
Turhan et al. [32]	1 leg	EVN 3 (osteomyelitis/swelling)	1	NA	Unknown	Amputation	Not mentioned	No
Iwao et al. [14]	Lower leg	EVN 3 (paralysis and swelling)	1	NA	Unknown	Shaving	Yes/stockings	No
Vaccaro et al. [33]	Lower leg	EVN type 3 (obesity; internal morbidity)	1	NA	Severe obesity	Antibiotics and died	Not mentioned	NA
Guarneri et al. [34]	Lower leg	EVN 3 (comorbidly)	1	Several	Unknown	Conservative without clearance of skin changes	Not mentioned	No
Freitas et al. [35]	Lower leg	EVN 3 (comorbidly)	1	Several	Unknown	Antibiotics and conservative treatment	Not mentioned	No
Eda et al. [36]	Lower leg	EVN 3 (comorbidly)	1	Several	BMI 44.2	Compression, obesity management, and tolvapan	Yes	No
Lee et al. [37]	Lower leg	EVN by scleroderma	1	Few	Unknown	Treatment of scleroderma	Not mentioned	No
Pitcher et al. [38]	Lower leg	EVN type 4	1	Many	BMI 27	Excision	Not mentioned	1 year

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
