# Peer review of "Shaving Technique and Compression Therapy for Elephantiasis Nostras Verrucosa (Lymphostatic Verrucosis) of Forefeet and Toes in End-Stage Primary Lymphedema: A 5 Year Follow-Up Study in 28 Patients and a Review of the Literature"

_jcm, 2020, doi:10.3390/jcm9103139_

Round 1

Reviewer 1 Report

The article analyzes a topic of extreme interest. The authors should broaden the discussion by considering therapeutic approaches described by other authors, such as systemic retinoids (Marasca C, Mascolo M, Ferrillo M, et al. Acitretin may improve symptoms and exudation in patients affected by elephantiasis nostras verrucosa: Report of a case. Int Wound J. 2019;16(2):579-581. doi:10.1111/iwj.13050).

Author Response

The article analyzes a topic of extreme interest. The authors should broaden the discussion by considering therapeutic approaches described by other authors, such as systemic retinoids (Marasca C, Mascolo M, Ferrillo M, et al. Acitretin may improve symptoms and exudation in patients affected by elephantiasis nostras verrucosa: Report of a case. Int Wound J. 2019;16(2):579-581. doi:10.1111/iwj.13050).

I want to thank the reviewer for the valuable comments.   In line 109 we mentioned already the treatment with retinoids with 1 publication.  We added your suggestion to it.

Reviewer 2 Report

This is an important report on long term outcomes in lymphedema management which are otherwise sparsely represented in the literature.

There are no major issues with the study, but I do have a major suggestion regarding terminology.

In lines 57 – 60, the authors describe the deterioration of lymphoedema and ENV as a late stage sequela of lymphatic failure, and in lines 91 – 93 use the term lymphostatic verrucosis. I suggest that since historical naming is a topic of the paper, that the authors take the opportunity to address and replace what can be considered outdated terms such as elephantiasis and notsrus. The former being associated with increased stigma for those affected, and the latter literally meaning 'us not them'. These words could easily be replaced with more accurately descriptive ones and I suggest reverting to the original as lymph stasis is a factor in all etiologies, and verrucosis has no undesirable connotations. Or at least use one of the others listed in line 92.

I also challenge the assertion that ENV needs to be distinguished by name from other forms of chronic lymphoedema involving significant papillomatous skin changes. Yes, causes of lymph statis must be investigated and appropriate treatment initiated, but the need to differentiate the resulting end points - which the authors state in line 48/49 are morphologically similar anyway, with different specific names, I think is over emphasized. Also, the reference in line 48 doesn’t really support the statement.  

Since the authors build a credible case for, and ultimately state in the conclusion that ENV is an end-stage of all the aforementioned causes of lymphatic failure, it seems a lost opportunity not to start out pointing to that more clearly in the introduction and suggesting usage of a more inclusive name.

Minor issues and suggestions

Line 35, the numerals 28/0 are redundant and a bit confusing, suggest removing.

Line 37, the BMI given as 29.4 does not match the data in Table 2

Line 44, I would argue the use of the term 'rare' here, unless it is used within a diagnostic context such as it is in ref 2. The condition itself is not uncommon in many populations affected by chronic edemas.

Line 48, I don’t agree that the given refence support the statement, and does not mention podoconiosis at all.

Line 56, refence to the ISL consensus document should be for the more recently published consensus of 2016

Line 73, suggest giving a brief distinction between cellulitis and erysipelas for readers not familiar with them and since both are used throughout the report.

Line 77 /78, the statement requires a reference to support the assertion that it is the therapist who finds toe bandaging too technically difficult. It could be argued that acceptability by patients is also a significant barrier to routine application as is accessibility.

Line 83, 86 90, the use of the term elephantiasis is erroneous here as it suggests this is a condition unique to tropical lymphedema when it is actually a descriptive name for advanced lymphedema presenting with visible pathogenic skin changes. It is correct to state that the largest cause of lymphoedema globally is lymphatic filariasis. Using 'advanced' or 'late stage' lymphedema instead of elephantiasis also makes more sense when the term non-filarial lymphoedema is used such as in the description of podoconiosis in line 84.

Line 95, the term toe caps needs a brief description for anyone unfamiliar with compression garments, and the statement made needs a supporting reference.

Table 2, the row 'Start swelling toes…' is not clear whether it is the age of onset or the duration of toe swelling. Is the triangle symbol used to infer 'mean'? It is more usually used as a symbol for change so should be explained in a footnote here. In the BMI data do the /x numerals as in 28/9 represent the n= in that BMI category? This table would benefit from some explanatory footnotes.

Line 211, the description of the study as large seems a little overstated, it is the first which should be emphasized.

Discussion, some of the most important topics are bit obscured by the lit review and I suggest reporting briefly on the review in the results, including moving the table of publications there. You can then tighten up the discussion to more clearly put forward the interpretation of your important findings.

Author Response

Reviewer 2

This is an important report on long term outcomes in lymphedema management which are otherwise sparsely represented in the literature.

There are no major issues with the study, but I do have a major suggestion regarding terminology.

In lines 57 – 60, the authors describe the deterioration of lymphoedema and ENV as a late stage sequela of lymphatic failure, and in lines 91 – 93 use the term lymphostatic verrucosis. I suggest that since historical naming is a topic of the paper, that the authors take the opportunity to address and replace what can be considered outdated terms such as elephantiasis and notsrus. The former being associated with increased stigma for those affected, and the latter literally meaning 'us not them'. These words could easily be replaced with more accurately descriptive ones and I suggest reverting to the original as lymph stasis is a factor in all etiologies, and verrucosis has no undesirable connotations. Or at least use one of the others listed in line 92.

I also challenge the assertion that ENV needs to be distinguished by name from other forms of chronic lymphoedema involving significant papillomatous skin changes. Yes, causes of lymph statis must be investigated and appropriate treatment initiated, but the need to differentiate the resulting end points - which the authors state in line 48/49 are morphologically similar anyway, with different specific names, I think is over emphasized. Also, the reference in line 48 doesn’t really support the statement.  

Since the authors build a credible case for, and ultimately state in the conclusion that ENV is an end-stage of all the aforementioned causes of lymphatic failure, it seems a lost opportunity not to start out pointing to that more clearly in the introduction and suggesting usage of a more inclusive name.

We thank you for the interessiting debate about the name,

Minor issues and suggestions

Line 35, the numerals 28/0 are redundant and a bit confusing, suggest removing.: Good suggestion: I changed it

Line 37, the BMI given as 29.4 does not match the data in Table 2: I corrected it

Line 44, I would argue the use of the term 'rare' here, unless it is used within a diagnostic context such as it is in ref 2. The condition itself is not uncommon in many populations affected by chronic edemas. Good point: I changed it.

Line 48, I don’t agree that the given refence support the statement, and does not mention podoconiosis at all. The reviewer is right: The text was confusing and I changed it.  Clinically the same; aetioology of the lymphostasis is different.

Line 56, refence to the ISL consensus document should be for the more recently published consensus of 2016: correct

Line 73, suggest giving a brief distinction between cellulitis and erysipelas for readers not familiar with them and since both are used throughout the report. I added some more information

Line 77 /78, the statement requires a reference to support the assertion that it is the therapist who finds toe bandaging too technically difficult. It could be argued that acceptability by patients is also a significant barrier to routine application as is accessibility.I rephrases the sentence.

Line 83, 86 90, the use of the term elephantiasis is erroneous here as it suggests this is a condition unique to tropical lymphedema when it is actually a descriptive name for advanced lymphedema presenting with visible pathogenic skin changes. It is correct to state that the largest cause of lymphoedema globally is lymphatic filariasis. Using 'advanced' or 'late stage' lymphedema instead of elephantiasis also makes more sense when the term non-filarial lymphoedema is used such as in the description of podoconiosis in line 84. I rephrased the sentence

Line 95, the term toe caps needs a brief description for anyone unfamiliar with compression garments, and the statement made needs a supporting reference. I added some more information

Table 2, the row 'Start swelling toes…' is not clear whether it is the age of onset or the duration of toe swelling. Is the triangle symbol used to infer 'mean'? It is more usually used as a symbol for change so should be explained in a footnote here. In the BMI data do the /x numerals as in 28/9 represent the n= in that BMI category? This table would benefit from some explanatory footnotes.: I changed the title and added the term duration for toes swelling. Further: I added the symbol “N”

Line 211, the description of the study as large seems a little overstated, it is the first which should be emphasized. I removed it

Discussion, some of the most important topics are bit obscured by the lit review and I suggest reporting briefly on the review in the results, including moving the table of publications there. You can then tighten up the discussion to more clearly put forward the interpretation of your important findings.
